# ROBOALIGN: REINFORCEMENT LEARNING FOR ACTION-ALIGNED MULTIMODAL LARGE LANGUAGE MODELS

## ABSTRACT

In recent years, state-of-the-art vision–language–action models (VLAs) have been built upon pre-trained multimodal large language models (MLLMs). However, how to systematically train MLLMs to improve VLA performance remains an open problem. While prior approaches primarily focus on strengthening embodied reasoning via linguistic actions, the modality gap limits the transferability of language-based knowledge to non-linguistic low-level actions produced by VLAs. To address this problem, we propose a novel framework ROBOALIGN that aligns MLLM representations with low-level actions, thereby producing MLLMs well-suited for VLA. Specifically, we achieve action alignment through reinforcement learning, where the model generates action tokens via zero-shot reasoning in natural language. To validate the effectiveness of ROBOALIGN, we train VLAs by adding a diffusion-based action head on top of an MLLM backbone and evaluate them on major robotics benchmarks. Specifically, training base MLLMs with ROBOALIGN improves the performance on robotic tasks by 17.5%, 18.9%, and 106.6% on LIBERO, CALVIN, and real-world robotic environments, respectively. Moreover, ROBOALIGN outperforms models aligned only with language-described actions or with supervised fine-tuning based approaches such as ECoT, demonstrating its effectiveness and broad applicability.

## 1 INTRODUCTION

Vision–language–action models (VLAs) have recently demonstrated remarkable success in robotics (Brohan et al., 2022; 2023; Driess et al., 2023). By integrating the visual perception, language understanding, and common-sense knowledge of multimodal large language models (MLLMs), VLAs provide a foundation for training generalizable robotic policies in real-world scenarios (Yang et al., 2023; Huang et al., 2022b; Tellex et al., 2020; Huang et al., 2022a; Hu et al., 2023). Specifically, policies are obtained either through discrete action token predictions by MLLMs (Kim et al., 2024; Pertsch et al., 2025; Kim et al., 2025b) or through continuous action prediction by external action experts that operate on latent states of MLLMs (Black et al., 2024; Bjorck et al., 2025; Team et al., 2024). This approach allows leveraging the extensive pretrained knowledge within MLLMs, enabling the development of generalizable policies even with a limited amount of robotics data.

However, the performance and generalization of VLAs are often limited by the underlying MLLMs, which struggle with key embodied tasks required for action generation, such as spatial reasoning (Tong et al., 2024; Zhou et al., 2025; Cheng et al., 2024) and temporal reasoning (Ahn et al., 2022; Sermanet et al., 2024). To address this limitation, researchers have developed various embodied question-answering tasks designed to improve reasoning skills for robotic manipulation. These include tasks such as answering high-level action questions (Chen et al., 2025; Lynch et al., 2023), responding to spatial questions about object relationships (Chen et al., 2024a; Xu et al., 2025), grounding points or bounding boxes in images to identify affordance-related locations (Yuan et al., 2024; Song et al., 2025a), and predicting future visual trajectories of end-effectors (Ji et al., 2025; Yuan et al., 2025a). While these tasks have been primarily addressed through supervised fine-tuning (SFT), recent approaches have applied reinforcement learning (RL) schemes (*e.g.*, DeepSeek-R1; Guo et al. 2025) to encourage reasoning, leading to significant improvements in performance (Azzolini et al., 2025; Kim et al., 2025a; Song et al., 2025b; Huang et al., 2025a).

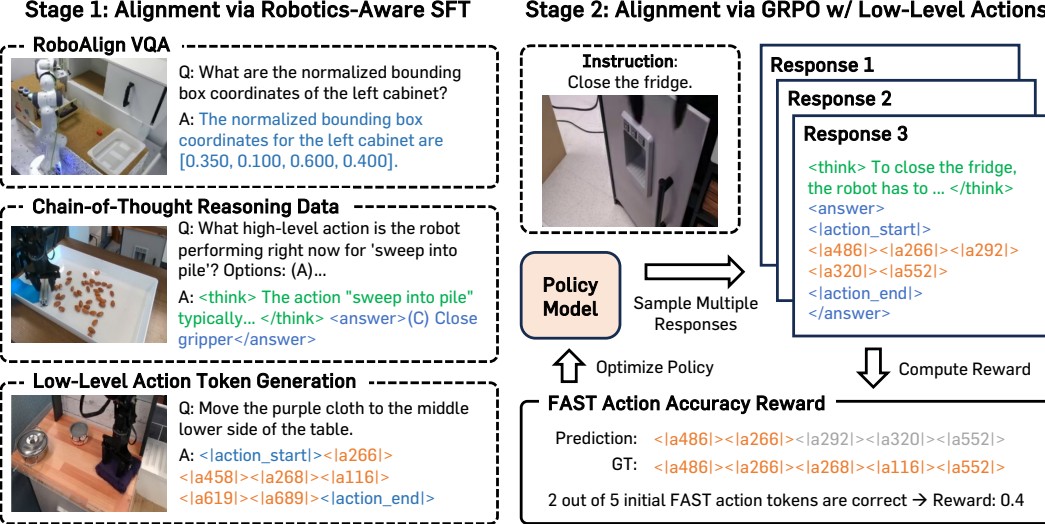

Figure 1: **Overview of ROBOALIGN framework.** ROBOALIGN directly aligns MLLM representations with low-level action generation using reasoning-incentivized reinforcement learning (Guo et al., 2025). The framework consists of two stages: (i) Stage 1 integrates embodied reasoning, zero-shot reasoning, and FAST-tokenized low-level action generation via supervised fine-tuning, and (ii) Stage 2 optimizes responses through reinforcement learning to improve token-level action accuracy and better alignment. The resulting model serves as an MLLM tailored for effective VLA training.

Despite recent successes in enhancing the embodied reasoning of MLLMs, *it remains unclear whether these improvements directly translate into improved low-level action generation in VLAs*, since language and low-level action modalities are inherently different and not naturally aligned. Moreover, such training is typically conducted through SFT, but it increases the risk of catastrophic forgetting (Chu et al., 2025), potentially weakening other capabilities of MLLMs essential for policy generation by VLAs. Motivated by this concern, we conducted experiments by training VLAs on top of open-source MLLMs specialized in embodied reasoning. Our experiments show that these specialized models indeed yield limited performance gains compared to the VLA model built upon the original, non-fine-tuned MLLM (see Figure 2).

**Contribution.** To address these limitations, we identify the necessity of aligning MLLMs directly using non-linguistic low-level actions. Motivated by this insight, we introduce ROBOALIGN, a training framework designed to directly align MLLM representations with low-level action generation, while coupling embodied reasoning capabilities with low-level actions.

The key idea of ROBOALIGN is an RL-based fine-tuning process that trains the MLLM to generate low-level action

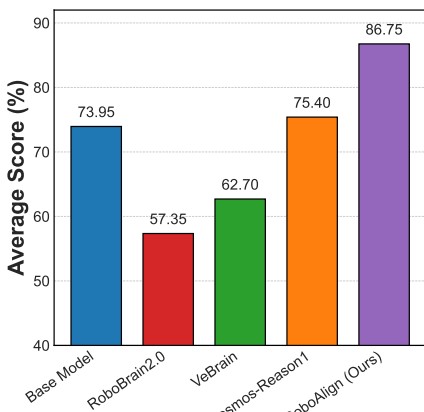

Figure 2: **Performance on LIBERO.** VLAs built upon MLLMs specialized for embodied reasoning (fine-tuned variants of Qwen2.5-VL-7B-Ins) fail to significantly improve performance and often degrade it compared to the baseline VLA based on the original model. In contrast, ROBOALIGN achieves significant gains, as detailed in Section 5.

tokens as the direct output of embodied reasoning. This allows the model to explore diverse embodied reasoning trajectories obtained through sampling and strengthens the coupling between reasoning and action generation, resulting in strong alignment between MLLM's internal knowledge and low-level actions. Moreover, this RL-based alignment reduces the risk of catastrophic forgetting compared to SFT, which is advantageous for preserving its general-purpose knowledge. Specifically, our method first fine-tunes the MLLM with SFT to enable the model to generate low-level actions through zero-shot reasoning, and then optimizes the model to further refine this reasoning process using GRPO (Shao et al., 2024) to maximize the action-accuracy reward.

To evaluate the effectiveness of RoboAlign, we train MLLMs with our framework and test the performance on a suite of robotic benchmarks, including simulation environments such as LIBERO (Liu et al., 2023) and CALVIN (Mees et al., 2022), as well as real-world robot settings. Specifically, we attach a diffusion-based action head to the frozen MLLM backbone and fine-tune it to generate low-level actions. Our experiments show that models trained with RoboAlign achieve substantial performance gains over the baseline models, with relative improvements of 17.5% on LIBERO, 18.9% on CALVIN, and 106.6% in the real-world setup. Moreover, we find that our approach is more effective than other alignment approaches such as high-level action prediction (13.1% v.s. 17.5%) or point trajectory prediction (15.2% v.s. 17.5%) on the LIBERO benchmark, respectively.

Furthermore, to examine if RoboAlign also improves embodied reasoning capabilities of MLLMs, we evaluated RoboAlign on a diverse set of benchmarks for general image understanding (Chen et al., 2024b), spatial reasoning (Song et al., 2025a; Yuan et al., 2024; Fu et al., 2024), and embodied reasoning for robotics (Kim et al., 2025a). On the embodied reasoning tasks, RoboAlign achieve state-of-the-art performance on embodied reasoning tasks, outperforming not only commercial general-purpose models such as GPT-4o (OpenAI, 2024), but also specialized embodied MLLMs, such as RoboBrain2.0 (Team et al., 2025). Notably, this is accomplished while preserving the model's performance on general image understanding. This result shows that our RL-based alignment enhances the general capabilities of MLLMs, in contrast to SFT-based alignment methods such as ECoT (Zawalski et al., 2024), which often degrades performance on these embodied tasks.

## 2 Related Work

**Multimodal large language models for robot control.** Efforts to leverage the visual processing capabilities, commonsense, and world knowledge of multimodal large language models (MLLMs) for robot policy decision have shown consistent success. In particular, MLLMs have demonstrated strong performance in high-level action planning. Concretely, prior work has explored generating predefined atomic action skills to directly control robots (Liang et al., 2023; Tellex et al., 2020; Luo et al., 2025), or producing high-level actions and plans that condition subsequent low-level actions (Driess et al., 2023; Yang et al., 2023; Huang et al., 2022b;a; Hu et al., 2023). These approaches have been further extended toward more precise action generation, either by enabling MLLMs to produce policies in an end-to-end manner (Kim et al., 2024; Pertsch et al., 2025; Kim et al., 2025b) or by training action experts that consume latent states instead of language outputs (Team et al., 2024; Li et al., 2023; Shentu et al., 2024; Black et al., 2024; Bjorck et al., 2025; GEAR, 2025). We investigate how to better align MLLMs with low-level actions to enhance such robot control performance.

**Multimodal large language model for embodied reasoning.** With the increasing application of MLLMs to embodied environments such as robot manipulation, their capabilities for tasks requiring spatial and temporal reasoning have been enhanced. For spatial reasoning, prior work has enhanced 3D scene understanding by leveraging VQA data to train models that convert information from 2D and 3D vision inputs (Chen et al., 2024a; Ray et al., 2024; Zhou et al., 2025; Wu et al., 2025). To further improve performance in specific robotic tasks, some approaches have trained models to predict bounding boxes or points associated with affordances and manipulation-relevant spatial cues (Yuan et al., 2024; Song et al., 2025a; Lu et al., 2023; Ji et al., 2025). For temporal reasoning, researchers have extracted high-level actions (Chen et al., 2025; Lynch et al., 2023; Chen et al., 2025; Huang et al., 2024; Chen et al., 2023), 2D point trajectories of object movement from egocentric videos of humans or robots to construct VQA (Huang et al., 2025a; Yang et al., 2025; Ranasinghe et al., 2024; Zheng et al., 2024; Lee et al., 2025). Nevertheless, these approaches only contribute indirectly to low-level action prediction.

**Encouraging reasoning through reinforcement learning.** Chain-of-Thought (CoT) prompting (Wang et al., 2022; Yao et al., 2023; Kim et al., 2023; Wei et al., 2022) has been widely applied to both LLMs and MLLMs in zero-shot, few-shot, and supervised fine-tuning (SFT) settings (Muennighoff et al., 2025), effectively improving answer quality. Recently, DeepSeek-R1 (Guo et al., 2025) proposed a training approach specialized for CoT, in which reasoning is explicitly enforced during the response process, and the entire reasoning trace is optimized using the reinforcement learning algorithm with rewards derived from the final answer. This training paradigm has demonstrated that, compared to SFT, models can achieve stronger performance and generalization across diverse domains, including mathematics (Zeng et al., 2025; Yu et al., 2025), agents (Lu et al., 2025; Jin et al.,

2025), visions (Shen et al., 2025; Huang et al., 2025b;b), and embodied reasoning (Kim et al., 2025a; Song et al., 2025b; Huang et al., 2025a; Yuan et al., 2025a;b), while requiring significantly less data, in some cases even a single example (Wang et al., 2025). In this work, we introduce a reinforcement learning scheme based on low-level action prediction, aligning the MLLM's representations more directly with robot control.

## 3 PRELIMINARIES

**FAST action tokenization.** We adopt FAST tokenization (Pertsch et al., 2025) to integrate low-level actions into MLLMs, as it has been shown to be effective not only for end-to-end policy learning but also for representation learning (Black et al., 2025; Driess et al., 2025). Our action is defined as a $D$-dimensional vector representing the end-effector's state, which consists of its Cartesian position $(x, y, z)$, orientation (roll, pitch, yaw), and gripper state (Open/Close). An action sequence over a horizon of $H$ timesteps forms a chunk, $\mathbf{a}_{1:H} = [[a_{1,1}, a_{1,2}, \ldots, a_{1,D}], \ldots, [a_{H,1}, a_{H,2}, \ldots, a_{H,D}]]$. To improve compactness, FAST tokenization transforms the action chunk $\mathbf{a}_{1:H}$ into the frequency domain using a discrete cosine transform (DCT; Ahmed et al. 2006). The resulting DCT coefficients are quantized and flattened into a sequence. This sequence is then compressed into discrete tokens using byte-pair encoding (BPE; Gage 1994), resulting in $T_k = \text{FAST}(\mathbf{a}_{1:H})$, where each token is mapped to one of $2K$ special tokens added to the MLLM's vocabulary for training and generation.

**Encouraging reasoning with GRPO.** To encourage explicit reasoning, we train the model to generate intermediate thoughts enclosed within `<think>...</think>` before producing a final answer. Training is conducted with Group Relative Policy Optimization (GRPO; Shao et al. 2024), where the policy is optimized jointly for format correctness and answer accuracy. Specifically, let the current policy be denoted as $\pi_{\theta_{\text{old}}}$. For a given query $q \sim P(Q)$, we sample $G$ responses $[o_1, \ldots, o_G] \sim \pi_{\theta_{\text{old}}}(q)$. Each response is evaluated by a pre-defined reward model $R(q, o_i)$, which assigns a reward $r_i$ based on format and answer accuracy. We then compute an advantage by normalizing the reward using the standard deviation, $A_i = \frac{r_i - \text{mean}(\mathbf{r})}{\text{std}(\mathbf{r})}$. GRPO optimizes the policy by maximizing these advantages while applying a KL penalty against a reference policy:

$$
\mathbb{J}_{\text{GRPO}}(\theta) = \mathbb{E}_{q \sim P(Q), \{o_i\}_{i=1}^{G} \sim \pi_{\theta_{\text{old}}}(\cdot|q)} \Bigg[
$$
$$
\frac{1}{G} \sum_{i=1}^{G} \min \left( \frac{\pi_\theta(o_i|q)}{\pi_{\theta_{\text{old}}}(o_i|q)} A_i, \text{clip} \left( \frac{\pi_\theta(o_i|q)}{\pi_{\theta_{\text{old}}}(o_i|q)}, 1 - \varepsilon, 1 + \varepsilon \right) A_i \right) - \beta \mathbb{D}_{\text{KL}}(\pi_\theta \| \pi_{\text{ref}}) \Bigg], \tag{1}
$$

where $\varepsilon$ and $\beta$ are hyperparameters for clipping and KL penalty.

## 4 ROBOALIGN: ALIGN EMBODIED REASONING WITH LOW-LEVEL ACTIONS

In this section, we introduce ROBOALIGN, a training framework that directly aligns multi-modal large language models (MLLMs) with low-level actions through reinforcement learning (RL). While doing so, ROBOALIGN is designed to preserve the general capabilities of MLLMs and simultaneously enhance embodied reasoning ability. A key challenge, however, is that off-the-shelf MLLMs cannot generate specialized low-level actions (*e.g.*, FAST tokens) in a zero-shot manner, making RL inapplicable. To address this, we introduce a two-stage training pipeline. First, we apply supervised fine-tuning (SFT) to equip the model with the initial ability to predict FAST tokens through zero-shot reasoning, while preserving the general abilities of MLLMs and enhancing embodied reasoning. Second, building on this ability, we apply RL on this SFT model to further strengthen embodied reasoning and improve FAST token prediction accuracy. The overall process is illustrated in Figure 1.

### 4.1 STAGE 1: INTEGRATING LOW-LEVEL ACTION WITH MLLM USING SFT

The primary objective of this SFT stage is to equip the MLLM with the ability to generate FAST action tokens, which is a prerequisite for the subsequent RL stage, while simultaneously preserving its general vision-language capabilities and enhancing its embodied reasoning skills. To achieve this, we curate a data mixture from four sources: (i) a variety of open-source SFT datasets for embodied

Table 1: **Example of the ROBOALIGN response.** Incorporating reasoning data during SFT effectively transfers zero-shot reasoning ability to FAST token generation process. Without such data, the model produces only minimal reasoning which reduces the diversity of reasoning samples and hinders RL.

**Question:** Your current task is 'pick up cup from the table'. Output the robot's actions to perform this task through Fast tokens.

**Model Answer (Trained w/ reasoning datasets):** <think>To pick up the cup from the table, the robot needs to move its gripper towards the cup, position it correctly, close the gripper to secure the cup, and then lift it up. Given the current state of the gripper being open and positioned above the cup, the immediate next action would be to move down towards the cup to prepare for grasping.</think><answer><|action_start|><|action_266|><|action_299|> ...

**Model Answer (Trained w/o reasoning datasets):** <think>Go to the cup.</think><answer><|action_start|><|action_266|><|action_299|>...

reasoning and general understanding, (ii) our custom ROBOALIGN VQA dataset for robotic embodied reasoning, (iii) specialized reasoning datasets designed to improve zero-shot reasoning of MLLMs, and (iv) robotic dataset with FAST tokens. We describe the process for building our custom datasets in this section, with full details for all data sources and configurations available in Appendix A.

**ROBOALIGN VQA.** While existing VQA datasets are useful for general embodied reasoning, high-quality VQA specifically grounded in robotic information remains limited. For example, datasets such as ShareRobot (Ji et al., 2025) and RoboVQA (Sermanet et al., 2024) use robot imagery but focus on high-level QA tasks, lacking the fine-grained, spatial-temporal information needed for low-level control. To address this gap, we develop a data generation pipeline that feeds robot images and associated metadata, *e.g.*, bounding boxes, end-effector states, and both high and low-level actions, into a powerful large model, *i.e.,* `gemini-2.5 pro` (Hassabis et al., 2025). The model then automatically generates a diverse set of high-quality VQA, captioning, and grounding QA pairs.

**Reasoning dataset with zero-shot CoT.** To preserve the MLLM's zero-shot reasoning ability during SFT and transfer it to the action generation process, we incorporate a specialized reasoning dataset into our training mixture. This dataset is created by distilling outputs from a reasoning model that is trained with GRPO to generate step-by-step reasoning. Specifically, we first train the reasoning model on spatial and robot-related embodied MCQAs for distillation, following Kim et al. (2025a). From this model, for a given prompt, we sample multiple reasoning outputs. These outputs are then filtered using a combination of rule-based rewards and correctness checks. Table 1 shows that including this specialized reasoning data during SFT enables the effective transfer of reasoning ability to FAST token generation, while the absence of such data results in limited zero-shot reasoning.

**FAST token generation dataset.** To enable FAST token prediction, we first extend the MLLM's vocabulary by adding two special marker tokens <ACTION_START>, <ACTION_END> and 2K FAST tokens. The training data is then constructed from the BridgeV2 dataset (Walke et al., 2023) in a QA format. Each sample pairs a robot image with a fixed instruction, where the ground-truth answer is the corresponding sequence of FAST tokens.

The resulting data mixture, consisting of our custom and open-source datasets, is used to fine-tune the MLLM with SFT, providing a strong foundation for subsequent RL training stage.

## 4.2 STAGE 2: ALIGNING EMBODIED REASONING WITH LOW-LEVEL ACTION USING RL

In the second stage, we use RL to directly align the MLLM with low-level actions, *i.e.,* FAST tokens, further refining the model to be better suitable for VLA adaptation. Specifically, we optimize the model's embodied reasoning process to directly improve the accuracy of FAST action token generation. To create the data for this stage, we adapt the FAST token dataset from Stage 1. In particular, each sample's input instruction is augmented with a prompt that requires explicit reasoning within <think>...</think> tags before producing the FAST token sequence.

We define the reward as the arithmetic mean of two components: a format reward $r_f \in \{0, 1\}$ indicating whether the output correctly adheres to the required reasoning format, and an accuracy reward $r_a \in [0, 1]$ measuring FAST token prediction accuracy. In particular, the accuracy reward $r_a$ is computed by measuring the prefix similarity between the generated action token sequence $T_{1:n}^{\text{gen}}$ and the target sequence $T_{1:m}^{\text{target}}$, normalized by the target length:

$$r_a = \frac{1}{m} \max\{i \in \{1, \ldots, m\} : T_{1:i}^{\text{gen}} = T_{1:i}^{\text{target}}\}. \tag{2}$$

The final reward is given by $r = (r_f + r_a)/2$. This formulation encourages the model to generate both correctly formatted and accurate FAST token sequences. Building on the constructed training dataset and reward function, we then apply GRPO (Shao et al., 2024) to further optimize the MLLM.

## 5 EXPERIMENT

In this section, we design experiments to answer the following research questions:

- ○ Does training with ROBOALIGN improve both MLLMs and the VLAs built upon them?
- ○ Is aligning with low-level actions more effective than alternative alignment methods?
- ○ Is RL-based alignment in ROBOALIGN more effective than SFT-based alignment?

### 5.1 EXPERIMENTAL SETUP

**Training data.** For supervised fine-tuning (SFT), we prepare a diverse set of datasets covering both general MLLM capability and Fast token prediction. In total, 1.88M samples are used for MLLM-related tasks. For FAST token prediction, we use the subset of BridgeV2 (Walke et al., 2023) dataset (400K samples), yielding 2.28M samples overall. For reinforcement learning (RL), we further use a 12.8K subset of the BridgeV2 FAST token prediction data. More details are provided in Appendix A.

**Baseline models.** To validate the effectiveness of ROBOALIGN, we prepare two baselines: (i) a model trained only on MLLM data and (ii) a model trained only on FAST token prediction using the full BridgeV2 dataset (1.88M samples). Both are trained for one epoch following the same SFT train schema as in ROBOALIGN.

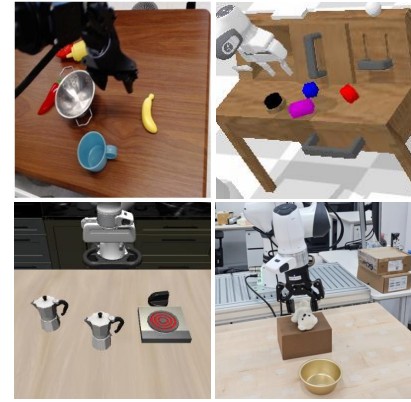

Figure 3: **Examples of Observations.** Visual inputs for training and evaluation (clockwise from top left): BridgeV2 for FAST token training, CALVIN, real-robot, and LIBERO benchmark.

**Benchmarks.** We evaluate VLA performance in LIBERO (Liu et al., 2023) and CALVIN (Mees et al., 2022) (see Figure 3 for the examples).

- **LIBERO:** This benchmark uses a Franka Panda Arm to perform manipulation tasks grouped into four categories: spatial, object, goal, and long-horizon. Each category consists of 10 tasks. Training uses the provided dataset covering all tasks, and evaluation runs 50 trials per task (500 trials per category).
- **CALVIN:** This benchmark also employs a Franka Panda Arm and consists of 34 distinct tasks. Training uses data collected from environments A, B, and C for 100K steps, after which zero-shot evaluation is performed in a novel environment D. Performance is measured by the success rate of executing five consecutive instruction chains, with a total of 1,000 chains evaluated.

**Implementation details.** We train our models based on Qwen2.5VL-7B-Ins (Bai et al., 2025). For SFT, we follow the official Qwen2.5VL training repository. The vision encoder is frozen, and we use a cosine scheduler with a learning rate of $2 \times 10^{-5}$, a warmup ratio of 0.03 and training for 1 epoch. For RL, we use the EasyR1 repository[1], training all parameters from scratch with a rollout batch size of 512, update batch size of 128, and 5 samples per prompt. We apply a constant learning rate of $1 \times 10^{-6}$ and train for one epoch. For VLA experiments, we adapt diffusion-based action head on

---

[1] https://github.com/hiyouga/EasyR1

Table 2: **LIBERO** success rates (%) for VLAs built upon MLLMs that were fine-tuned with various methods, evaluated over 500 trials per category. Each model is evaluated by training a newly-initialized, diffusion-based action head on the LIBERO dataset while the MLLM backbone remains frozen. ROBOALIGN shows particularly large improvements in the Long and Goal categories compared to other training methods.

| Method | Spatial | Object | Goal | Long | Avg. |
|---|---|---|---|---|---|
| Qwen2.5VL-7B-Ins (Bai et al., 2025) | 95.2 | 95.0 | 42.4 | 63.2 | 73.9 |
| w/ Language-Only SFT | 91.0 | 94.4 | 67.8 | 65.0 | 79.6 |
| w/ Action-Only SFT | 89.8 | 95.8 | 82.8 | 57.6 | 81.5 |
| w/ ROBOALIGN (SFT) | 92.8 | 97.4 | 59.0 | 65.6 | 78.7 |
| w/ ROBOALIGN (SFT + RL) | 93.8 | 96.0 | 87.2 | 70.0 | **86.8** |

Table 3: **CALVIN ABC→D** success rates (%) for VLAs built upon MLLMs that were fine-tuned with various methods, evaluated over 1000 trials. Each model is evaluated by training a newly-initialized, diffusion-based action head on the CALVIN dataset while the MLLM backbone remains frozen. While all baselines show drops in task completions of length 4 and 5, ROBOALIGN consistently improves performance across all sequence.

| Method | Task completed in a row (%) ↑ | | | | | Succ. Len. |
|---|---|---|---|---|---|---|
| | 1 | 2 | 3 | 4 | 5 | (Avg) |
| Qwen2.5VL-7B-Ins (Bai et al., 2025) | 77.8 | 55.0 | 38.6 | 26.6 | 18.1 | 2.16 |
| w/ Language-Only SFT | 87.4 | 62.2 | 41.9 | 25.2 | 15.3 | 2.32 |
| w/ Action-Only SFT | 66.1 | 34.7 | 15.3 | 7.1 | 3.2 | 1.26 |
| w/ ROBOALIGN (SFT) | 74.6 | 49.6 | 31.5 | 21.2 | 12.2 | 1.89 |
| w/ ROBOALIGN (SFT+RL) | 87.6 | 67.2 | 47.1 | 32.8 | 22.2 | **2.57** |

top of an MLLM backbone and train newly-initialized diffusion-based action head on robot datasets while keeping the MLLM backbone frozen. Action experts are newly trained for each benchmark environment with a batch size of 32. Training steps are set to 60K for LIBERO, 100K for CALVIN (see detail in Appendix A)

## 5.2 MAIN RESULTS

As shown in Tables 2, 3, MLLMs trained with ROBOALIGN, which combines SFT and RL, achieve the highest performance across all simulations. The SFT stage alone yields moderate improvements, suggesting that most of the performance gain comes from the RL stage. In particular, ROBOALIGN demonstrates a significant increase in success rates on long-horizon tasks, which are more intricate and complex than other types of tasks. For example, in CALVIN (Table 3), ROBOALIGN achieves the highest task completions of length-5 success rate (18.1% → 22.2%), whereas all other training methods show a decline performance in here. Similarly, in LIBERO (Table 2), the *Long* category improves to 70% with ROBOALIGN, compared to only ~2% gains from other methods.

Another notable finding is in the *Goal* category of LIBERO, which requires handling different instruc-

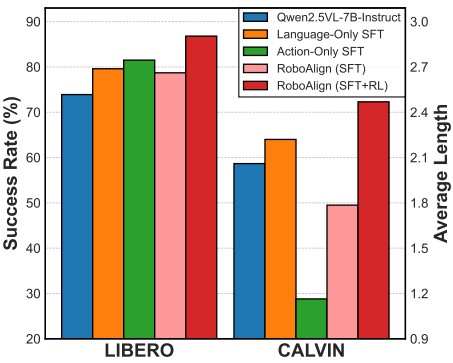

Figure 4: **Summary of VLA performance.** Comparison of VLA performance across different MLLM training methods on LIBERO and CALVIN. ROBOALIGN achieves the highest gains across all settings.

tions in the same environment. Here, ROBOALIGN improves performance dramatically from 42.4% to 87.2%. However, models trained only with MLLM data show limited improvements. Specifically, in CALVIN they achieve higher success in task completions of length-1 (77.8% → 87.4%) but experience a drop in task completions of length-5 performance (18.1% → 15.3%). Similarly, in LIBERO they improve in the *Goal* category (42.4% → 67.8%) but yield only marginal gains in the *Long* category (63.2% → 65.6%). These results indicate that embodied reasoning abilities learned

Table 4: **Real robot** success rates (%) for VLAs built upon MLLMs that were fine-tuned with various methods, evaluated over 96 trials per task. Each model is evaluated by training a newly-initialized, diffusion-based action head on the real-world robotic dataset while the MLLM backbone remains frozen. We find that ROBOALIGN is also effective in real-world settings.

| Method/ stage | Box to bowl | Box to plate | Basket to bowl | Plate to basket | Avg. |
|---|---|---|---|---|---|
| Qwen2.5VL-7B-Ins (Bai et al., 2025) | 16.7 | 70.8 | 20.8 | 20.8 | 32.3 |
| w/ ROBOALIGN (SFT) | 87.5 | 58.3 | 37.5 | 37.5 | 55.2 |
| w/ ROBOALIGN (SFT+RL) | 87.5 | 58.3 | 70.8 | 50.0 | **66.7** |

Table 5: **Compatibility with different models.** We apply ROBOALIGN to a different MLLM backbone (Qwen3VL-8B-Ins) to validate its generalizability. We report success rates (%) on the LIBERO benchmark, averaged over 500 trials per category. ROBOALIGN consistently improves overall performance, with particularly significant gains in the Long category.

| Method | Spatial | Object | Goal | Long | Avg. |
|---|---|---|---|---|---|
| Qwen3VL-8B-Ins (Team, 2025) | 94.2 | 96.4 | 90.0 | 60.0 | 85.2 |
| w/ ROBOALIGN (SFT) | 96.2 | 97.4 | 93.3 | 71.0 | 89.5 |
| w/ ROBOALIGN (SFT + RL) | 95.6 | 99.6 | 95.2 | 78.6 | **92.5** |

through language can enhance performance on relatively simple tasks, but offer limited improvements on more complex and demanding tasks. When trained only with VLA data, we observe large in-domain gains, particularly in LIBERO's *Goal* category (42.4% → 82.8). However, performance drops significantly on long-horizon tasks in both CALVIN and LIBERO. We hypothesize that while FAST token training strengthens alignment between instructions and low-level actions in-domain, it also induces forgetting of general MLLM capabilities, leading to reduced zero-shot generalization.

## 5.3 ABLATION STUDY AND ANALYSES

**Real robot experiments.** To examine whether the improvements of ROBOALIGN on VLA performance extend beyond simulation to real-robot settings, we conduct experiments using a Franka Research 3 robot arm across four distinct pick-and-place tasks. Each task involves moving a different object (teddy bear, box, cup, sponge). Training is performed with 60 demonstrations per task, and evaluation consists of 24 trials per object, totaling 96 trials per task. The VLA setup follows the same configuration as in the main experiments, with each task trained for 30K steps. As shown in Table 4, ROBOALIGN consistently improves performance even in real-robot settings.

**Compatibility with different models.** To assess whether ROBOALIGN generalizes to other architectures, we conducted experiments using another MLLM backbone, Qwen3-VL-8B-Ins. For the MLLM training phase, we utilized 5K samples for RL, while maintaining all other training setup. After training, all models are converted into VLAs and evaluated on the LIBERO simulation environments. As shown in Table 5, we observed an overall performance increase, with particularly significant gains in the Long category. This trend is consistent with the results observed in Table 2. These results demonstrate that ROBOALIGN effectively generalizes across different MLLM architectures.

**Comparison with embodied alignment strategies.** To evaluate the effectiveness of aligning with low-level action by ROBOALIGN, we compare it with two commonly used embodied MLLM training tasks: (i) predicting high-level actions expressed in language descriptions and (ii) predicting 2D visual trajectories of the end effector. For a fair comparison, all models are trained with RL on the same BridgeV2 images as ROBOALIGN, encouraging embodied reasoning in both cases. For high-level action alignment, we convert movements such as "*move right*" or "*move left*" into multiple-choice QA format and provide rewards based on correctness (Kim et al., 2025a). For 2D visual trajectory prediction, we use data from ShareRobot and adopt the same reward formulation as ThinkAct (Huang et al., 2025a). Since ShareRobot contains only 6K samples, training is limited to this size. After training, all models are converted into VLAs and evaluated on the LIBERO simulation environments. As shown in Table 6, ROBOALIGN achieves the largest performance improvement. In contrast, the alternative methods show notable gains in the LIBERO *Goal* category but remain limited on long-horizon tasks. This trend is consistent with our main results in Section 5.3 and further demonstrates the advantage of direct alignment with low-level actions.

Table 6: **Impact of alignment strategies on VLA.** We compare different RL alignment strategies on the LIBERO benchmark, reporting success rates (%), evaluated over 500 trials per category. All experiments start from the same SFT model and use an identical RL setup, with only the alignment target varying between experiments. ROBOALIGN consistently improves performance and uniquely enhances long-horizon tasks, where other methods degrade.

| Method | Spatial | Object | Goal | Long | Avg. |
|---|---|---|---|---|---|
| ROBOALIGN (SFT) | 92.6 | 97.4 | 65.2 | 64.0 | 79.1 |
| w/ Language-described high-level action alignment | 91.6 | 94.6 | 90.0 | 58.2 | 83.6 |
| w/ Robot 2d point trajectory forecasting alignment | 92.4 | 95.6 | 87.8 | 64.6 | 85.1 |
| w/ Low-level Action alignment (Ours) | 93.8 | 96.0 | 87.2 | 70.0 | **86.8** |

Table 7: **Comparison with SFT-based alignment.** We compare our RL-based alignment against an SFT-based baseline that jointly trains reasoning and low-level actions using the ECoT (Zawalski et al., 2024) dataset. Both methods are fine-tuned from the ROBOALIGN SFT model and evaluated on the LIBERO benchmark, reporting success rates (%), evaluated over 500 trials per category. While the SFT-based baseline degrades performance, ROBOALIGN achieves significant improvements.

| Method | Spatial | Object | Goal | Long | Avg. |
|---|---|---|---|---|---|
| ROBOALIGN (SFT) | 92.6 | 97.4 | 65.2 | 64.0 | 79.1 |
| w/ SFT-based Alignment (ECoT) | 84.6 | 90.8 | 49.6 | 45.6 | 67.7 |
| w/ RL-based Alignment (Ours) | 93.8 | 96.0 | 87.2 | 70.0 | **86.8** |

**Comparison with SFT-based alignment.** We further compare RL–based alignment in ROBOALIGN against SFT-based alignment. Specifically, we consider ECoT (Zawalski et al., 2024), which aligns reasoning and low-level actions through SFT. For this experiment, we use the ECoT dataset while keeping the action space in the form of FAST tokens. Both methods are trained on the same 12.8K samples on top of the ROBOALIGN SFT model, with one epoch of SFT using identical hyperparameters. Then, the resulting models are converted into VLAs and evaluated on the LIBERO simulation environments. As shown in Table 7, the SFT-based approach even reduces performance compared to RL. We attribute this to the limited generalization of SFT, where knowledge aligned on BridgeV2 transfers poorly to LIBERO, as well as to forgetting effects introduced during SFT. Consistently, when evaluated on general MLLM benchmarks, the ECoT-trained model shows a degradation in performance, confirming the limitations of SFT-based alignment.

**Performance on MLLM benchmarks.** To examine whether ROBOALIGN enhances embodied reasoning and generalist capabilities of MLLM, we evaluate performance across diverse MLLM benchmarks. We use MMStar (Chen et al., 2024b) for general VQA ability, Robospatial-Home (Song et al., 2025a), Where2Place (Yuan et al., 2024), and the depth components of BLINK (Fu et al., 2024) for spatial reasoning. For robot embodied reasoning, we use Robot-R1 Bench (Kim et al., 2025a), which provides detailed assessments of embodied reasoning abilities including planning, subtask decomposition, movement, and spatial reasoning, all based on BridgeV2. As shown in Table 8, ROBOALIGN outperforms specialized embodied reasoning models such as Cosmos-Reason1 (Azzolini et al., 2025), RoboBrain2.0 (Team et al., 2025), and VeBrain (Luo et al., 2025) across embodied reasoning tasks, while maintaining strong performance on general MLLM benchmarks. In contrast, Cosmos-Reason1 and RoboBrain2.0 show clear drops in general task performance. Furthermore, RL–based alignment with low-level actions does not reduce MLLM capability, but instead improves it. We attribute this to the alignment of embodied reasoning with low-level action generation, which simultaneously enhances both action accuracy and embodied reasoning performance.

# 6 CONCLUSION

We proposed ROBOALIGN, a training framework for multimodal large language models (MLLMs) tailored to vision–language–action models (VLA) by directly aligning MLLM's representations with low-level action policies. Our approach leverages reinforcement learning to improve low-level action prediction accuracy through embodied reasoning. We evaluated ROBOALIGN across diverse robotic environments and MLLM benchmarks, and demonstrated that it consistently delivers substantial gains in embodied reasoning performance within MLLM tasks as well as in the VLA domain across

Table 8: **Performance on multimodal benchmarks.** We evaluate ROBOALIGN and other MLLMs on general image understanding (MMStar), spatial reasoning (RoboSpatial, Where2Place, BLINK, and robot embodied reasoning (Robot-R1 Bench) benchmarks. Our initial SFT model, ROBOALIGN (SFT), performs on par with specialized embodied-reasoning MLLMs, and RL training further boosts performance across the overall MLLM benchmarks. Values marked with ∗ are taken from prior work (Team et al., 2025; Duan et al., 2024).

| Model | MMStar | Robot-R1 Bench (0-3) | RoboSpatial | Where2Place | Blink (Rel. Depth) |
|---|---|---|---|---|---|
| GPT-4o-2024-11-20 (Hurst et al., 2024) | 65.10* | 1.55 | 44.42* | 20.41* | 77.90* |
| Qwen2.5-VL-7B-Ins (Bai et al., 2025) | 60.30 | 1.02 | 36.29 | 11.35 | 55.64 |
| Cosmos-Reason1-7B (Azzolini et al., 2025) | 54.40 | 1.19 | 38.81* | 5.51* | 68.57* |
| RoboBrain2.0-7B (Team et al., 2025) | 35.80 | 1.17 | **54.23*** | **63.59*** | 83.95* |
| VeBrain-8B (Luo et al., 2025) | 61.90 | 1.02 | 42.48* | 11.34* | 79.68* |
| ROBOALIGN (SFT) | 62.47 | 1.14 | 48.86 | 51.66 | 87.10 |
| ROBOALIGN (SFT+RL) | **62.80** | **1.38** | 50.86 | 54.49 | **87.90** |

both short and long horizon tasks. In contrast, language-only embodied reasoning fine-tuning yields limited or even degraded performance on complex scenarios. These results establish ROBOALIGN as an effective and generalizable approach for advancing VLA training.

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

# A    EXPERIMENT DETAILS

## A.1    COMPUTING COST

We use 8×H200 GPUs for MLLM training, requiring approximately 30 hours for SFT and 1 hour for reinforcement learning. For VLA training, we use 2×A100 GPUs, with each 10K training steps taking about 1 hour of computation.

## A.2    IMPLEMENTATION DETAILS FOR VLA TRAINING

Our implementation refers to the GR00T-N1.5 codebase[2] (GEAR, 2025), adopting the same architecture with an initialized diffusion policy action expert. The expert takes as input the hidden states from the 18-th layer of Qwen2.5VL-7B-Ins (Bai et al., 2025). Hyperparameters for policy fine-tuning follow those of the official GR00T-N1.5 implementation unless otherwise specified.

## A.3    TRAINING DATASETS

For supervised fine-tuning (SFT), we prepare a diverse set of datasets covering both general MLLM capability and embodied reasoning. To preserve general multimodal ability, we use 100K samples from LLaVA-OneVision (single-view only) (Li et al., 2024). For embodied reasoning, we include 300K samples from RefSpatial (Zhou et al., 2025), 200K from RoboPoint (Yuan et al., 2024), 50K from EgoPlan-IT (Chen et al., 2023), and 500K from our own multi-view instruction dataset. To enhance robot-specific embodied reasoning, we incorporate 100K samples each from ShareRobot (Ji et al., 2025) and RobotVQA (Sermanet et al., 2024), 150K from our RoboAlign VQA, and 300K from BridgeV2 (Walke et al., 2023) and Droid (Khazatsky et al., 2024) Robot QA (predicting movements such as "move right," "move left," the current 7-DoF state, and a future sequence of 10 states). Since conventional robot imitation environments do not take video inputs, video-based datasets (RobotVQA, EgoPlan-IT, ShareRobot) are converted into single-frame inputs by extracting the last frame. For reasoning data, we include 50K multiple-choice QA samples converted from our RoboAlign VQA dataset and another 50K derived from SAM2 (Ravi et al., 2024), which queries spatial relations among key objects. Of these, 30K samples are used to train the reasoning distillation model. After augmenting with generated data and applying correctness filtering, the final reasoning dataset consists of 76K samples. In total, the MLLM training set contains 1.88M QA samples. For FAST token prediction (Pertsch et al., 2025), we use the subset of BridgeV2 dataset (400K samples). For reinforcement learning, we further use a 12.8K subset of the BridgeV2 FAST token prediction data. The training for FAST token prediction follows the prompt template shown in Figure 5.

---

**Waypoint prediction QA for SFT**

**System Prompt**

You are an embodied vision-language robotic assistant for multi-object manipulation.

**System Prompt for Reinforcement Learning**

You are an embodied vision-language robotic assistant for multi-object manipulation. The assistant first thinks about the reasoning process in the mind and then provides the user with the answer. The reasoning process and answer are enclosed within <think> </think> and <answer> </answer> tags, respectively.

**Prompt**

Your current task is instruction. Output the robot's actions to perform this task through Fast tokens.

---

Figure 5: **Prompt for FAST Token generation** We use this prompt template for both FAST token prediction and reinforcement learning.

---

[2]https://github.com/NVIDIA/Isaac-GR00T

Table 9: **K-Nearest Neighbor Accuracy.** We measure how accurately MLLM representations can predict underlying states using KNN classification on 20 trajectories from a LIBERO task.

| Method | Acc. (%) |
|---|---|
| Qwen3VL-8B-Ins (Team, 2025) | 39.06 |
| w/ ROBOALIGN (SFT) | 43.23 |
| w/ ROBOALIGN (SFT + RL) | **69.79** |

## B ADDITIONAL ANAYLSIS

### B.1 K-NEAREST NEIGHBORHOOD BASED REPRESENTATION ANALYSIS

In this section, we analyze how ROBOALIGN affects the underlying MLLM representations. We hypothesize that explicit aligning low-level actions enables the model to learn more discriminative and fine-grained features for action generation. To evaluate this, we perform a KNN classification experiment that measures how accurately the MLLM representation can predict similar underlying states. We select 20 training trajectories from one of the LIBERO long-horizon tasks, "put the white mug on the left plate and put the yellow and white mug on the right plate." We assign each timestep to 32 classes using Dynamic Time Warping (DTW) (Müller, 2007) over robot states. We then evaluate whether the MLLM, receiving only vision and task instruction, can recover the correct underlying class using a KNN classifier ($k = 5$) applied to its hidden representation. As shown in Table 9, ROBOALIGN (SFT+RL) produces substantially more discriminative representations than both baselines, improving KNN accuracy from $39.06\%$ to $69.79\%$. This result indicates that the RL alignment stage significantly sharpens the model's ability to encode fine-grained state information. Distinct representation help to generate accurate actions, and these results help to understand the mechanism of ROBOALIGN's performance improvement.

### B.2 RL TRAINING PROCESS

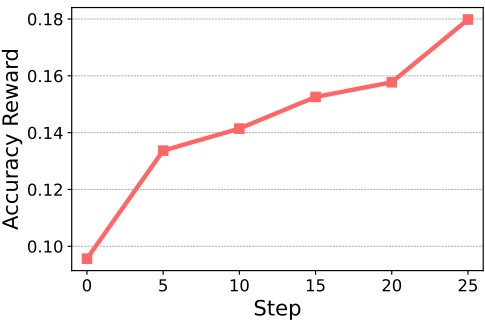
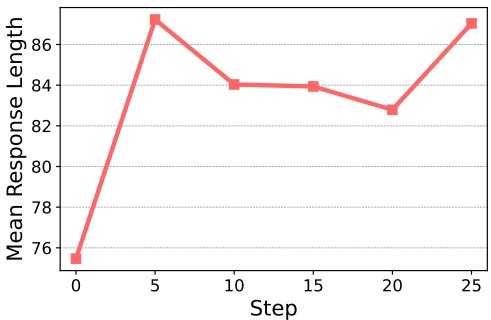

(a) Accuracy reward change during RL training  (b) Response length change during RL training

Figure 6: **Training metrics for reinforcement learning in ROBOALIGN.** Validation performance during RL training: (a) accuracy reward increases steadily, while (b) response length remains stable.

Figures 6 and 2 illustrate how the average response length and accuracy reward evolve during the RL training of ROBOALIGN. The average response length remains stable, while the accuracy reward gradually increases, indicating effective learning.

Table 7 provides qualitative examples before and after RL training. Prior to training, responses mainly describe high-level plans with minimal detail on immediate actions (e.g., "the robot needs to first approach the cloth"). After RL training, the model generates more concrete action descriptions, such as "the robot needs to first approach and align its gripper with the cloth. Once aligned, it will need to close the gripper to pick up the cloth.". In addition, a novel element of situational awareness has been introduced (e.g., "The robot's current position suggests it is already aligned with the cloth, ready

to pick it up."). These quantitative and qualitative results together demonstrate that RL training is successfully improving the model's ability to align reasoning with actionable steps.

---

**Question:** Your current task is move the orange cloth to the top of the right burner. Output the robot's actions to perform this task through Fast tokens.

---

**GT Answer:**
<|action_start|><|action_486|><|action_265|><|action_268|><|action_116|><|action_269|><|action_end|>

---

**ROBOALIGN (SFT) Answer:**
'<think>To perform the task of moving the orange cloth to the top of the right burner, the robot needs to first approach the cloth, then grasp it, and finally move it to the right burner. The actions should be sequential and purposeful, focusing on the cloth and the burner in question.</think><answer>
<|action_start|><|action_266|><|action_709|><|action_268|><|action_116|><|action_269|><|action_end|>

---

Reward: 0.5 (Format 1.0 / Accuracy 0.0)

---

**ROBOALIGN (SFT+RL) Answer:** <think>To perform the task of moving the orange cloth to the top of the right burner, the robot needs to first approach and align its gripper with the cloth. Once aligned, it will need to close the gripper to pick up the cloth, lift it, and then move it to the right burner before releasing it. The robot's current position suggests it is already aligned with the cloth, ready to pick it up.</think><answer>
<|action_start|><|action_486|><|action_265|><|action_268|><|action_116|><|action_269|><|action_end|>

---

Reward: 1.0 (Format 1.0 / Accuracy 1.0)

---

Figure 7: **Example of model response**

## C   USE OF AI TOOLS

We acknowledge that a large language model (LLM) was used to refine the phrasing and grammar of the manuscript.

