# OpenReview forum: "RoboAlign: Reinforcement Learning for Action-Aligned Multimodal Large Language Models"
_ICLR.cc/2026/Conference — Submitted to ICLR 2026_

### Official Review · Reviewer_8H9S · 2025-10-29

**Soundness:** 2
**Presentation:** 3
**Contribution:** 2
**Rating:** 4
**Confidence:** 4

**Summary:**

The paper proposes ROBOALIGN, a reinforcement learning–based training framework that aligns multimodal large language models (MLLMs) with low-level action generation for embodied manipulation. The method aims to strengthen the connection between high-level reasoning and executable actions while retaining the model’s general-purpose knowledge. ROBOALIGN first applies supervised fine-tuning (SFT) to let the model generate low-level actions via zero-shot reasoning, and then employs GRPO optimization to refine this process through action-accuracy rewards. This RL-based alignment encourages diverse embodied reasoning trajectories, improves reasoning-action coupling, and mitigates catastrophic forgetting compared with SFT alone.

**Strengths:**

1. In my view, the proposed approach effectively reduces downstream training costs (e.g., on LIBERO and CALVIN) by freezing the MLLM backbone and training only the action head. This design is, in my opinion, the most significant contribution of the paper.

2. The authors conduct both simulation and real-robot experiments, which substantially enhance the credibility and practicality of the results.

3. The idea of using GRPO-based alignment to strengthen the MLLM’s capability in low-level action generation is novel and insightful.

**Weaknesses:**

1. As mentioned in the strengths, the most valuable contribution of this work is its ability to reduce downstream training costs. However, the paper does not clearly emphasize or analyze this aspect. From the experimental results alone, the performance on LIBERO is still lower than many recent approaches, such as the representative work OpenVLA-OFT [1].

2. The practical impact of this contribution appears limited. In current downstream VLA training scenarios, freezing the VLA backbone provides only minor time savings, while achieving higher task success rates remains the main focus. Therefore, the proposed method may lack strong practical relevance or real-world applicability.

3. The paper provides limited details and analysis regarding the experimental setup and results. It would benefit from a more comprehensive discussion or additional analyses in the appendix to help readers better understand the experimental design and the effectiveness of the proposed method. (Suggestion, not affect rating.)

[1] Fine-Tuning Vision-Language-Action Models: Optimizing Speed and Success.

**Questions:**

1. During the GRPO experiments, did the authors observe genuine improvements in the model’s robotic generalization ability? Specifically, can the model trained on Bridge V2 fast-token data truly enhance its capacity to generate accurate low-level actions?

2. In the SFT stage, the authors fine-tune the MLLM on several reasoning datasets. Are these datasets truly meaningful for improving downstream embodied tasks such as LIBERO and CALVIN? I am skeptical about their practical value, as such reasoning data may have limited impact while increasing training cost. It would be helpful if the authors could provide ablation studies to validate this.

Minor Issues / Typos:

1. It is recommended that all equations include numbering for easier reference and citation.

2. In line 276, within the reward formulation, the index i should start from 1 rather than 0.

---

> ### Author Response · Authors · 2025-11-22
> **Response to Reviewer 8H9S [1/2]**
>
> Dear Reviewer 8H9S,
>
> We sincerely appreciate your constructive feedback. We have carefully addressed each of your questions and provided detailed responses below. Major revisions in the revised manuscript are highlighted in blue.
>
> ---
>
> **[Correction of our research scope]**
>
> We first emphasize that our research does not focus on the VLA training algorithm itself (e.g., training a VLA module on top of a frozen VLM). Instead, our primary goal is to investigate how to pre-train a VLM backbone that yields optimal representations for downstream VLA training (i.e., when VLA is trained on top of a pretrained VLM). Therefore, our experimental setup fixes the VLA training methodology to the proven gr00tN1.5 approach, measuring the performance differences resulting from various VLM pre-training strategies.
>
> ---
>
> **[Correction of our contribution]**
>
> Our core contribution/novelty lies in newly introducing non-linguistic low-level actions for the VLM pre-training stage, while most existing studies have relied on linguistic actions (which are more friendly understandable by VLM), e.g., image coordinates related to robot [1] or high-level actions [2], which limits the transferability to non-linguistic low-level action generation of VLA due to the modality gap. To address this limitation, we propose a novel framework that optimizes the model based on low-level actions directly, while empirically demonstrating that this approach is significantly more effective than existing methods.
>
> Furthermore, we observe that merely training models to generate low-level actions yields limited alignment effects (see Tables 2 and 3). To overcome this, we propose a method that explicitly aligns the VLM's general capabilities with low-level actions by simultaneously coupling them with the VLM's embodied reasoning. To enable this, we employ reasoning-incentivized RL.
>
> However, implementing this is non-trivial, as applying reasoning-incentivized RL to tokens across different modalities remains underexplored. We address this challenge by naturally transferring zero-shot reasoning ability to low-level actions through joint SFT with zero-shot embodied reasoning QA and introducing a novel reward formula.
>
> In summary, our work focuses not on the construction of VLA models themselves, but rather on investigating how to pre-train a VLM backbone specifically tailored for VLA training. Therefore, our primary contribution is the proposal of a novel VLM pre-training approach and a specialized framework. We have revised the Abstract and Introduction sections to further emphasize the unique contributions of ours.
>
> ---
>
> **[W1,2] Why not clearly emphasize or analyze VLA training method, and show low performance compared to other VLA**
>
> As previously mentioned, our research focuses on pre-training a VLM backbone to get suitable representations for VLA training, rather than on the VLA model architecture or training recipe itself. Thus, we fixed the VLA training method by utilizing the established GR00T N1.5 framework to assess different pre-training strategies. Consequently, since the VLA training process itself lies outside our contribution.
>
> Furthermore, a direct comparison with other VLA architectures, such as OpenVLA-OFT, is unsuitable. These frameworks operate at a different stage of the pipeline and are therefore orthogonal to our focus on VLM pre-training. For example, OpenVLA-OFT can be trained on top of a VLM trained with RoboAlign. This shows that direct comparison with other VLA learning methodologies is inappropriate.
>
> ---
>
> **[W3] Additional analysis**
>
> We highlight that the current manuscript already provides detailed experimental setups. And we also conducted various analyses regarding the VLM training methodology, specifically:
> - Data Composition for SFT: Analysis of language data, action data, and co-training effects (Tables 2 and 3).
> - RL Optimized Target: Performance changes when optimizing for different objectives, such as language-described high-level actions vs. robot 2D point trajectory forecasting (Table 5).
> - Alignment Methods: Comparison between SFT-based coupling embodied reasoning with low level action and our RL-based approach (Table 6).
>
> Please let us know if you need any further details or analysis.

---

> ### Author Response · Authors · 2025-11-22
> **Response to Reviewer 8H9S [2/2]**
>
> **[Q1] Observation of robotic generalization ability**
>
> During the GRPO experiments, did the authors observe genuine improvements in the model’s robotic generalization ability? Specifically, can the model trained on Bridge V2 fast-token data truly enhance its capacity to generate accurate low-level actions?
>
> We observed significant improvements in generalization across diverse robotic environments. Our experimental setup involves pre-training the VLM solely on the BridgeV2 dataset (collected with a WidowX robot) and subsequently converting it into a VLA for evaluation across distinct environments: CALVIN (Franka Panda), LIBERO (Franka Panda), and a Real Robot setup (Franka Research 3). These environments differ in camera viewpoints, robot embodiments, and tasks.
>
> Despite these domain gaps, applying GRPO consistently yielded substantial performance gains compared to the baseline across all environments. Notably, as shown in Table 3 (CALVIN), while SFT on robot data initially led to performance degradation, applying GRPO successfully reversed this trend, achieving significant generalization and performance improvements. These results empirically demonstrate the strong generalization capability of the RoboAlign.
>
> ---
>
> **[Q2] Effect of reasoning datasets**
>
> The inclusion of reasoning datasets is a critical prerequisite for enabling the Reinforcement Learning process, which is the most important factor in enhancing RoboAlign's performance.  As shown in Table 1, without these reasoning datasets, the model fails to generate diverse or sufficiently long reasoning traces and struggles to adhere to the required format. As a result, reinforcement learning itself becomes impossible, making it impossible to perform RoboAlign itself.
>
> Regarding training costs, the reasoning dataset comprises only 30K samples out of a total 1.88M (approximately 1.6%), representing a negligible fraction of the data mixture. Consequently, the computational cost added by incorporating this data is minimal compared to the essential capability it unlocks for the Roboalign framework.
>
> ---
>
> **[Minor Issues / Typos]**
>
> Thank you for your suggestion, we will work to fix it.
>
> ---
>
> [1] Huang et al., Thinkact: Vision-Language-Action Reasoning via Reinforced Visual Latent Planning, Arxiv 2025
>
> [2] Azzolini et al., Cosmos-reason1: From physical common sense to embodied reasoning, Arxiv 2025
>
> ---
>
> If you have any further questions/concerns, please do not hesitate to let us know.
>
> Thank you very much,
>
> Authors

---

> ### Comment · Reviewer_8H9S · 2025-11-24
> **Good response, but not enough**
>
> Thanks for authors positive response!
>
> - Regarding W1 & W2:
>
> I understand the authors’ intention: the SFT+GRPO pipeline is meant to enhance the underlying VLM. However, the contribution of this training recipe is still limited, and more importantly, the paper does not analyze whether this procedure actually benefits downstream VLA applications in practice.
>
> A key concern is practicality: after training a VLM with SFT+GRPO, would plugging it into mainstream VLA systems (e.g., OpenVLA) and then performing the standard end-to-end VLA training still yield meaningful improvements? Note that systems like OpenVLA do not train “VLA on a frozen VLM”—they optimize the entire model jointly. It is unclear whether the proposed SFT+GRPO stage provides advantages under such realistic training setups.
>
> - Regarding W3:
>
> When I asked for analysis, I meant explaining the reasons behind the observed experimental results, not simply restating the numbers. For aspects that are difficult to analyze directly, reasonable hypotheses are still expected. Currently, some of the analyses in the paper—especially in the Main Results section—are not very informative, as they mainly restate the outcomes without deeper interpretation.
>
> If the authors can adequately address the above issues, I would be willing to raise my score.

---

> ### Author Response · Authors · 2025-11-28
> **Response to Reviewer 8H9S**
>
> **[W1 & W2] Standard end-to-end VLA**
>
> The reviewer pointed out that standard end-to-end VLA training differs from "VLA on a frozen VLM." However, we believe that our current setup, freezing the VLM and training a diffusion action expert, is actually closer to the latest standard practice. Current SOTA commercial models such as GR00T-N1.5[1] and Pi0.5[2] train with gradients from the diffusion action expert blocked. This approach helps to accelerate training speed and preserve the VLM's capabilities as a generalist and even show high performance.
>
> Nevertheless, we understand concern, so we conducted additional experiments to validate the effectiveness in a setup where the VLM is also fine-tuned. We train VLA on LIBERO with VLM gradients enabled under the same VLA training environment. But we observe  the loss decreased only to 0.2, compared to 0.06 in our original setup, indicating that policy learning barely occurred. We attribute this is because our VLA training setup uses only fine-tuning data without pre-training. This result suggests that training the VLM jointly requires significantly larger-scale data and computational resources. Despite our best efforts, we were unable to obtain results under this setup within our current resources.
>
> Meanwhile, the VLM functions as a backbone encoder, and across most vision-language research (classification, segmentation, robotics, etc.), when performance is strong with a frozen encoder, unfreezing the encoder generally follows the same trend. Therefore, we expect that RoboAlign would show similar performance improvements even in setups where the VLM is trained jointly.
>
> ---
>
> **[W3] Additional Analysis**
>
> Thank you for clarifying question. We will provide additional interpretation further analysis of experimental results based on your questions.
>
> The core contribution of this paper is explicitly aligning low-level actions with the VLM's representation. This enables the representation to be learned with consideration for precise movements that are difficult to express in language. In contrast, models trained only with language or not strongly aligned with low-level actions cannot provide sharp representations for individual fine-grained actions. Distinct representations have a significant impact on accurately generating actions, and we believe this is where RoboAlign's performance gains lie.
>
> To further support our hypothesis, we conducted an additional analysis using KNN classification to examine whether the VLM produces distinguishable representations for complex tasks. Specifically, for one of the LIBERO Long tasks, “put the white mug on the left plate and put the yellow and white mug on the right plate”, we collected 20 training trajectories and mapped each individual timestep into one of 32 classes using Dynamic Time Warping (DTW) [3] based on the underlying real robot state.
>
> We then evaluated whether the VLM representation, which only receives vision and task instruction as input, can correctly predict the class corresponding to similar underlying states using a KNN classifier.
>
> \begin{array}{c|ccccc}
> \hline
> \text{Model} & \text{KNN accuarcy} \newline \hline
> \text{QwenVL-3} & 39.06 &  \newline \hline
> \text{RoboAlign (SFT)} &  43.23 & \newline \hline
> \text{RoboAlign (SFT+RL)} & 69.79  \newline\hline
> \end{array}
>
> As shown in the table above, the RoboAlign (SFT+RL) model produces substantially more discriminative representations than both the baseline. This indicates that RoboAlign’s RL stage significantly sharpens the VLM’s ability to encode fine-grained information, further supporting our claim that aligning low-level actions with representation learning is crucial for performance increase.
>
> We will add this discussion to the Appendix.B section.
>
> ---
> [1] NVIDIA Research, “GR00T-N1.5: Embodied Foundation Model for Robotics,” NVIDIA Research Project Page, 2024. \
> [2] Intelligence, Physical, et al. “$\pi_{0.5}$: A Vision-Language-Action Model with Open-World Generalization.” arXiv preprint arXiv:2504.16054, 2025. \
> [3] Müller, Dynamic time warping, Information Retrieval for Music and Motion, 2007
>
> ---
>
> If you have any further questions/concerns, please do not hesitate to let us know.
>
> Thank you very much,  \
> Authors

---

### Official Review · Reviewer_aMPf · 2025-11-01

**Soundness:** 2
**Presentation:** 3
**Contribution:** 2
**Rating:** 4
**Confidence:** 3

**Summary:**

The authors present RoboAlign, a training method for VLAs that aims to address the difficulty of translating improvements in MLLM quality to low-level action generation. RoboAlign consists first of a supervised fine-tuning stage, in which the MLLM is trained to generate FAST action tokens using a curated mixture of VQA and MCQA datasets formatted specifically to encourage reasoning. Then RoboAlign performs a reinforcement learning fine-tuning stage (GRPO), in which low-level action tokens output are encouraged to match the target tokens for as long as possible in the sequence. The method is implemented on top of the Qwen2.5VL-7B-Ins MLLM and experiments are performed on Libero and Calvin to test whether RoboAlign improves over typical training. On Libero, RoboAlign is compared to two other action alignment strategies.

**Strengths:**

- Method design allows many VLA architectures.
- Interesting and promising performance shown on multimodal benchmarks.

**Weaknesses:**

- Only one MLLM backbone is used in the experiments. The results would generalize if improvements could be shown with another MLLM.
- Baselines do not include another VLA such as GR00T N1 [1] or OpenVLA [2].

[1] GR00T N1: An Open Foundation Model for Generalist Humanoid Robots. Bjorck et al., arXiv preprint arXiv:2503.14734 2025.
[2] OpenVLA: An Open-Source Vision-Language-Action Model. Kim et al., PMLR, 2025.

**Questions:**

- Does RoboAlign work only for VLAs that generate action tokens? Certain works such as LLaRA [1] or TraceVLA [2] output actions in image coordinate space.
l. 185 How long does sampling take? How do you compute the reward based on the response?
Table 3: What is the difference between language-only and action-only SFT?
Section 5.3 Real Robot Experiments: How do other baselines perform on these real world tasks?

[1] LLaRA: Supercharging Robot Learning Data for Vision-Language Policy. Li et al., ICLR 2025.
[2] TraceVLA: Visual Trace Prompting Enhances Spatial-Temporal Awareness for Generalist Robotic Policies. Zheng et al., CoRL 2024.

---

> ### Author Response · Authors · 2025-11-22
> **Response to Reviewer aMPf [1/2]**
>
> Dear Reviewer aMPf,
>
> We sincerely appreciate your constructive feedback. We have carefully addressed each of your questions and provided detailed responses below. Major revisions in the revised manuscript are highlighted in blue.
>
> ---
>
> **[W1] other backbone MLLM**
>
> We conducted experiments solely on Qwen2.5-VL, since we believe it possesses sufficient representativeness in the building pipelines of recent VLAs. In this year, Qwen2.5-VL not only demonstrates high performance but has also become the most widely used model due to its active open-source support. Furthermore, since numerous studies on embodied reasoning [1;2] and VLA [3;4] have been conducted solely based on Qwen2.5-VL, we considered it to have strong representativeness in this field.
>
> However, we agree with the reviewer's suggestion that validation on other models is necessary. To address this, we conducted new experiments using Qwen3VL-8B which has a different architecture and training data distribution compared to the Qwen2.5-VL. For evaluation, we train VLA using the same way in the paper and evaluate on LIBERO simulation benchmark. The results are as follows:
>
> \begin{array}{c|ccccc}
> \hline
> \text{LIBERO} & \text{Long} & \text{Goal} & \text{Object} & \text{Spatial} & \text{Avg.} \newline
> \hline
> \text{QwenVL-3} & 60.0 & 90.0 & 96.4 & 94.2 & 85.2 \newline
> \hline
> \text{RoboAlign (SFT)} & 71.0 & 93.3 & 97.4 & 96.2 & 89.5 \newline
> \hline
> \text{RoboAlign (SFT+RL)} & 78.6 & 95.2 & 99.6 & 95.6 & 92.5 \newline\hline
> \end{array}
>
> The results show that there is an overall performance increase, with a particularly large performance improvement in the long subsection. These results demonstrate that Roboalign effectively generalizes across different VLM architectures. We report the results in Section 5.3 in the revision.
>
> ---
>
> **[W2] Baselines do not include another VLA**
>
> We clarify that our research focuses on pre-training a VLM backbone to ensure optimal representations for VLA training, not on the VLA model architecture or training recipe itself. Thus, we fixed the VLA training method in our experiments to evaluate pre-training strategies.
>
> For this reason, VLA training frameworks such as, GR00T N1, OpenVLA, are orthogonal to our contribution, since that represent a different stage of the pipeline. For example, OpenVLA and Gr00T-N1 can be trained on top of a VLM trained with RoboAlign.
>
> This shows that direct comparison with other VLA learning methodologies is inappropriate.
>
> ---
>
> **[Q1] RoboAlign can output actions in image coordinate space?**
>
> RoboAlign works for other types of actions as well. Since, the fundamental idea of RoboAlign is optimizing the model for action generation by coupling actions with embodied reasoning and aligning action with VLM’s representation. This alignment strategy is agnostic to the specific action.
>
> Furthermore, [1; 5] have already demonstrated that models can effectively handle image coordinates for reasoning and response generation. For this reason, we attribute that extending RoboAlign to such coordinate-based methods would be straightforward.
>
> ---
>
> **[Q2] l. 185 How long does sampling take?**
>
> Sampling is highly efficient. On a setup with 4x H200 GPUs, generating 500 responses takes approximately 1 second, achieving a generation throughput of around 4,000 tokens/s.
>
> ---
>
> **[Q3] How do you compute the reward based on the response?**
>
> Please refer to Section 4.2 (Line 274) for the detailed reward formulation. Briefly, we calculate the accuracy reward based on the length of the matching prefix between the generated action token sequence and the ground truth sequence.

---

> ### Author Response · Authors · 2025-11-22
> **Response to Reviewer aMPf [2/2]**
>
> **[Q4] What is the difference between language-only and action-only SFT?**
>
> Language-Only SFT: This refers to pre-training the VLM using only text-based embodied reasoning datasets. It focuses on enhancing the model's spatial and temporal reasoning capabilities through natural language tasks, without exposing the model to low-level action tokens.
>
> Action-Only SFT: This refers to pre-training the VLM using only robot action generation data. In this setting, the model is trained to predict low-level action tokens (FAST tokens) directly from task prompts.
>
> ---
>
> **[Q5] How do other baselines perform on these real world tasks?**
>
> As mentioned in [W2], baselines such as GR00T N1 and OpenVLA are orthogonal to our research; therefore, a direct comparison is unsuitable.
>
> ---
>
> [1] Huang et al., Thinkact: Vision-Language-Action Reasoning via Reinforced Visual Latent Planning, Arxiv 2025
>
> [2] Hung et al., Nora: A small open-sourced generalist vision language action model for embodied tasks, Arxiv 2025
>
> [3] Azzolini et al., Cosmos-reason1: From physical common sense to embodied reasoning, Arxiv 2025
>
> [4] Kim et al., Robot-R1: Reinforcement Learning for Enhanced Embodied Reasoning in Robotics, Arxiv 2025
>
> [5] Sarch et al., Grounded Reinforcement Learning for Visual Reasoning, Arxiv 2025
>
> ---
>
> If you have any further questions/concerns, please do not hesitate to let us know.
> Thank you very much,
>
> Authors

---

> ### Author Response · Authors · 2025-11-28
> **Gentle Reminder**
>
> Dear Reviewer aMPf,
>
> We once again thank you for your review of our manuscript. We sincerely appreciate your valuable insights and feedback. As the discussion period is drawing to a close, we kindly remind you that there are five days remaining for further comments or questions. We would be grateful for the opportunity to address any additional concerns you may have before the discussion phase ends.
>
> Thank you for your time and consideration.
>
> Warm regards,\
> Authors

---

### Official Review · Reviewer_nTe5 · 2025-11-01

**Soundness:** 2
**Presentation:** 3
**Contribution:** 2
**Rating:** 2
**Confidence:** 3

**Summary:**

This paper introduces RoboAlign, a framework designed to better align MLLM representations with low-level action generation in a VLA. The key contribution is applying GRPO, a reinforcement learning method, after the standard SFT stage to enhance policy alignment and control performance. Experiments using Qwen2.5VL-7B-Instruct demonstrate that RoboAlign improves robot rollout success across popular benchmarks like CALVIN and LIBERO, as well as in real-world robotic settings.

**Strengths:**

This paper is clearly presented and well-organized, which makes it easy to follow. The authors covered all the necessary preliminary knowledge to understand the paper, and the experiments cover both two well-known simulation benchmarks and real-world environments.

**Weaknesses:**

The core contribution of this paper is to empirically find that applying GRPO could be helpful for VLAs (or MLLMs) and further improves the performance after SFT. However, since GRPO itself is not a novel method and its application to a VLA—especially one adapted from a VLM—is relatively intuitive, the work’s conceptual novelty is somewhat limited. Nevertheless, the demonstration of its practical success in robotics is still valuable.

The key limitation that contributes a lot to the review is that the proposed approach is evaluated only on a single VLM, Qwen2.5VL-7B-Instruct, making it unclear whether the observed improvements generalize to other base models or architectures. The authors are encouraged to validate the generality of their method by testing it with additional models; otherwise, the paper is more similar to a tech report instead of a research paper.

**Questions:**

N/A

---

> ### Author Response · Authors · 2025-11-22
> **Response to Reviewer nTe5**
>
> Dear Reviewer nTe5,
>
> We sincerely appreciate your constructive feedback. We have carefully addressed each of your questions and provided detailed responses below. Major revisions in the revised manuscript are highlighted in blue.
>
> ---
>
> **[W1]  Lack of contribution**
>
> We clarify our research does not aim to train a specific robot policy via RL, nor is it merely an empirical attempt to enhance SFT performance.  We focus on how to pre-train a VLM backbone that yields optimal representations for VLA training (i.e., when VLA is trained on top of a pretrained VLM).
>
> We acknowledge that applying RL to enhance performance in specific domains is a well-established concept. However, the novelty of our work does not stem from the RL algorithm itself; rather, our novel contribution lies in identifying the necessity of aligning VLMs directly using non-linguistic low-level actions. While existing studies typically solve the problem by enhancing embodied reasoning via linguistic actions, e.g., image coordinates related to robot [1] or high-level actions [2], we demonstrate that the modality gap limits the transferability of language-based knowledge of VLM to non-linguistic low-level actions produced by VLA.
>
> Furthermore, we observed that merely training models to generate low-level actions yields limited alignment effects (see Tables 2 and 3). To address this, we propose a method that explicitly aligns the VLM's general capabilities with low-level actions by simultaneously coupling them with the VLM's embodied reasoning. To enable this, we employ reasoning-incentivized RL.
>
> However, implementing this approach is non-trivial, as applying reasoning-incentivized RL to tokens across different modalities remains underexplored. We address this challenge by naturally transferring zero-shot reasoning ability to low-level actions through joint SFT with zero-shot embodied reasoning QA and introducing a novel reward formula.
>
> To sum up, our work is not merely an empirical performance improvement achieved via RL, but rather a proposal of a novel VLM pre-training approach and a specialized framework. We have revised the Abstract and Introduction sections to further emphasize the unique contributions of ours.
>
> ---
>
>
> **[W2] Other base model**
>
> We conducted experiments solely on Qwen2.5-VL, since we believe it possesses sufficient representativeness in the building pipelines of recent VLAs. In this year, Qwen2.5-VL not only demonstrates high performance but has also become the most widely used model due to its active open-source support. Furthermore, since numerous studies on embodied reasoning [2,3] and VLA [1,4] have been conducted solely based on Qwen2.5-VL, we considered it to have strong representativeness in this field.
>
> However, we agree with the reviewer's suggestion that validation on other models is necessary. To address this, we conducted new experiments using QwenVL3-8B-Instruct which has a different architecture and training data distribution compared to the Qwen2.5-VL. For evaluation, we train VLA using the same way in the paper and evaluate on LIBERO simulation benchmark. The results are as follows:
>
> \begin{array}{c|ccccc}
> \hline
> \text{LIBERO} & \text{Long} & \text{Goal} & \text{Object} & \text{Spatial} & \text{Avg.} \newline
> \hline
> \text{QwenVL-3} & 60.0 & 90.0 & 96.4 & 94.2 & 85.2 \newline
> \hline
> \text{RoboAlign (SFT)} & 71.0 & 93.3 & 97.4 & 96.2 & 89.5 \newline
> \hline
> \text{RoboAlign (SFT+RL)} & 78.6 & 95.2 & 99.6 & 95.6 & 92.5 \newline\hline
> \end{array}
>
> The results show that there is an overall performance increase, with a particularly large performance improvement in the long subsection. These results demonstrate that Roboalign effectively generalizes across different VLM architectures. We report the results in 5.3 section in the revision.
>
> ---
>
> [1] Huang et al., Thinkact: Vision-Language-Action Reasoning via Reinforced Visual Latent Planning, Arxiv 2025
>
> [2] Azzolini et al., Cosmos-reason1: From physical common sense to embodied reasoning, Arxiv 2025
>
> [3] Kim et al., Robot-R1: Reinforcement Learning for Enhanced Embodied Reasoning in Robotics, Arxiv 2025
>
> [4] Hung et al., Nora: A small open-sourced generalist vision language action model for embodied tasks, Arxiv 2025
>
> ---
> If you have any further questions/concerns, please do not hesitate to let us know.
> Thank you very much,
>
> Authors

---

> ### Author Response · Authors · 2025-11-28
> **Gentle Reminder**
>
> Dear Reviewer nTe5,
>
> We once again thank you for your review of our manuscript. We sincerely appreciate your valuable insights and feedback. As the discussion period is drawing to a close, we kindly remind you that there are five days remaining for further comments or questions. We would be grateful for the opportunity to address any additional concerns you may have before the discussion phase ends.
>
> Thank you for your time and consideration.
>
> Warm regards,\
> Authors

---

### Official Review · Reviewer_u9JW · 2025-11-01

**Soundness:** 3
**Presentation:** 4
**Contribution:** 2
**Rating:** 6
**Confidence:** 5

**Summary:**

This paper presents ROBOALIGN: a system for directly aligning multimodal LLMs (MLLM) with lower-level action generation via a two phase training pipeline:
1- It uses supervised fine-tuning (SFT) to equip the model with the initial ability to predict action tokens.
2- It uses reasoning-incentivized reinforcement learning (following other work like Deepseek-R1) improve performance and reduce problems like catastrophic forgetting.

The SFT phase is done on a data mixture that was curated from various VQA and other CoT datasets, while the RL is using offline GRPO with a subset of the SFT CoT data. FAST is used to tokenize the actions into tokens.

The paper tests ROBOALIGN on a couple of robotic simulation benchmarks (CALVIN, LIBERO) as well as real-world robots. Showing uplift.

**Strengths:**

- The paper is well-presented and easy to read.

- The paper shows that ROBOALIGN training process keeps the baseline performance on multimodal benchmarks or improves it.

- The paper reports success rates on a real Franka Research 3 robot arm. This sets it apart from other work that just tests on simulations. And it shows significant uplift against the baseline Qwen model.

**Weaknesses:**

- Many of the observations/design decisions that the paper makes are now considered the standard for action models (i.e. reasonable action tokenization from continuous vectors to discrete tokens, choosing a reasonable data mixture and doing some kind of SFT first, then some kind of RL -potentially with CoT- as a final step). The real contribution is not in the full pipeline in my opinion since it seems to be similar to previous works. The real contributions is the combination of algorithmic choices (e.g. FAST for tokenization, GRPO for RL), and their own curated data mixture including "ROBOALIGN VQA" dataset.

- The ROBOALIGN training pipeline uses only offline RL with repeated data from the SFT. It does not use any online RL training (e.g. with gym/mujoco simulation environments or similar).

- While in some evaluation cases the improvements are significant, some evaluations show degradation (especially on long-horizon tasks)

- More ablations studies and evaluations would be interesting in my opinion and would make the contribution stronger, examples:
-- Other base models apart from Qwen-2.5 to see if these findings translate to other baselines
-- Two benchmarks are good but the work could benefit from including more benchmarks in both training and evaluation (e.g. metaworld)

**Questions:**

Typos:
- line 101: "of is embodied"

---

> ### Author Response · Authors · 2025-11-22
> **Response to Reviewer u9JW [1/2]**
>
> Dear Reviewer u9JW,
>
> We sincerely appreciate your constructive feedback. We have carefully addressed each of your questions and provided detailed responses below. Major revisions in the revised manuscript are highlighted in blue.
>
> ---
>
> **[w1] Lack of contribution**
>
> We first emphasize that our research focuses on how to pre-train a VLM backbone that yields optimal representations for VLA training (i.e., when VLA is trained on top of a pretrained VLM).
>
> To this end, we align the VLM with actions using an SFT--RL pipeline, as the reviewer noted. Here, our core contribution/novelty lies in newly introducing non-linguistic low-level actions for the VLM pre-training stage, while most existing studies have relied on linguistic actions (that are more friendly understandable by VLM), e.g., image coordinates related to robot [1] or high-level actions [2], which limits the transferability to non-linguistic low-level action generation of VLA due to the modality gap. To address this limitation, we propose a novel framework that optimizes the model based on low-level actions directly, while empirically demonstrating that this approach is significantly more effective than existing methods.
>
> However, implementing this approach is non-trivial, as applying reasoning-incentivized RL to tokens across different modalities has been underexplored. We address this challenge by naturally transferring zero-shot reasoning ability to low-level actions through joint SFT with zero-shot embodied reasoning QA and introducing a novel reward formula.
>
> Overall, our work is not merely a combination of existing algorithms, but rather a proposal of a novel VLM training approach and a specialized methodology. We have revised the Abstract and Introduction sections to further emphasize the unique contributions of ours.
>
> ---
>
> **[W2] Without online learning**
>
> As clarified in [W1] response, we employ RL not to optimize a robot action policy, but to align the VLM to learn representations suitable for VLA training.
>
> For these reasons, the primary goal is training the VLM itself. The "policy" in our context corresponds to the LLM's logits, distinct from a robot action policy. The VLM performs sampling of reasoning and action tokens for each new prompt, and we optimize the entire generated response via on-policy RL, guided by rewards provided by an external function. The mentioned online RL involving simulation interaction is related to robot policy that would be therefore orthogonal to our current research focus.
>
> While incorporating interaction with a simulation environment could potentially aid in acquiring higher-quality data, it introduces significant trade-offs. Specifically, it incurs additional computational costs due to environmental interaction. Furthermore, the introduction of a new RL objective increases framework complexity, which could potentially lead to unforeseen optimization issues.
>
> ---
>
> **[W3] Performance Degradation (especially on long-horizon tasks)**
>
> We think this is your misunderstanding. As shown in Tables 2, 3, 4, 5, and 6, we emphasize that our method demonstrates almost no degradation; notably, it achieves the highest performance across all cases for the mentioned long-horizon tasks. As emphasized in our response to [W1], since the RL stage is the key component of our framework, "RoboAlign (SFT+RL)" represents the complete proposed method, and the evaluation should be based on this configuration.

---

> ### Author Response · Authors · 2025-11-22
> **Response to Reviewer u9JW [2/2]**
>
> **[W4] Other base model + Evaluation**
>
> We conducted experiments solely on Qwen2.5-VL, since we believe it possesses sufficient representativeness in the building pipelines of recent VLAs. In this year, Qwen2.5-VL not only demonstrates high performance but has also become the most widely used model due to its active open-source support. Furthermore, since numerous studies on embodied reasoning [2,3] and VLA [1,4] have been conducted solely based on Qwen2.5-VL, we considered it to have strong representativeness in this field.
>
> However, we agree with the reviewer's suggestion that validation on other models is necessary. To address this, we conducted new experiments using QwenVL3-8B-Instruct which has a different architecture and training data distribution compared to the Qwen2.5-VL. For evaluation, we train VLA using the same way in the paper and evaluate on LIBERO simulation benchmark. The results are as follows:
>
> \begin{array}{c|ccccc}
> \hline
> \text{LIBERO} & \text{Long} & \text{Goal} & \text{Object} & \text{Spatial} & \text{Avg.} \newline
> \hline
> \text{QwenVL-3} & 60.0 & 90.0 & 96.4 & 94.2 & 85.2 \newline
> \hline
> \text{RoboAlign (SFT)} & 71.0 & 93.3 & 97.4 & 96.2 & 89.5 \newline
> \hline
> \text{RoboAlign (SFT+RL)} & 78.6 & 95.2 & 99.6 & 95.6 & 92.5 \newline\hline
> \end{array}
>
> The results show that there is an overall performance increase, with a particularly large performance improvement in the long subsection. These show Roboalign effectively generalizes across different VLM architectures.  We report the results in 5.3 section in the revision.
>
> Regarding the additional benchmark experiments, we are currently preparing them and will do our best to share the results within the discussion period. Meanwhile, given that we have already verified generalization capabilities across three distinct environments (LIBERO, CALVIN, and real-robot). We are confident that our method will readily generalize to the other environments as well.
>
> ---
>
> **[Minor Issues / Typos]**
>
> Thank you for your suggestion. We have fixed the typo and reflected it in the new revision.
>
> ---
>
> [1] Huang et al., Thinkact: Vision-language-action reasoning via reinforced visual latent planning, Arxiv 2025
>
> [2] Azzolini et al., Cosmos-reason1: From physical common sense to embodied reasoning, Arxiv 2025
>
> [3] Kim et al., Robot-R1: Reinforcement Learning for Enhanced Embodied Reasoning in Robotics, Arxiv 2025
>
> [4] Hung et al., Nora: A small open-sourced generalist vision language action model for embodied tasks, Arxiv 2025
>
> ---
> If you have any further questions/concerns, please do not hesitate to let us know.
>
> Thank you very much,  \
> Authors

---

> > ### Author Response · Authors · 2025-11-28
> > **Additional Response to Reviewer u9JW**
> >
> > **[W4] Other Evaluation**
> >
> > We have already verified generalization capabilities across three distinct environments (LIBERO, CALVIN, and real-robot). RoboAlign consistently showed performance improvements across these benchmarks, which we believe sufficiently demonstrates the generalizability of our approach.
> >
> > However, as the reviewer suggested, adding further benchmarks can strengthen our argument. Therefore, we additionally conducted evaluations on Metaworld. For VLA training, we trained each task for 10K steps, covering six diverse tasks of varying difficulty levels. The results are shown below:
> >
> > \begin{array}{c|ccccc}
> > \hline
> > \text{Metaworld} & \text{Press Button} & \text{Reach} & \text{Box Close} & \text{Coffee Push}  & \text{Lever Pull}  & \text{Soccer}  & \text{Avg.} \newline
> > \hline
> > \text{QwenVL-3} & 0.96 &	0.08 &	0 &	0.76 &	0.06 &	0.18 & 	0.34 \newline
> > \hline
> > \text{RoboAlign (SFT)} & 1 &	0.32 &	0.06 &	0.84 &	0.16 &	0.12 &	0.42 \newline
> > \hline
> > \text{RoboAlign (SFT+RL)} &1 &	0.38 &	0.08 &	0.84 &	0.42 &	0.24 &	0.49 \newline\hline
> > \end{array}
> >
> > The results show that the RoboAlign model achieves a clear overall performance increase compared to the baseline, further confirming its effectiveness and generalization capability.
> >
> > ---
> >
> > If you have any further questions/concerns, please do not hesitate to let us know.
> >
> > Thank you very much,
> > Authors

---

### Author Response · Authors · 2025-11-22
**General Response**

Dear Reviewers and Area Chair,

We sincerely appreciate your time and effort in serving the ICLR community.

We address an important issue in the field of pretrain VLM for VLA which is a newly identifying the necessity of aligning VLMs directly using non-linguistic low-level actions. To this end, we propose a novel framework to align MLLM representations with low-level actions using RL (8H9S). This approach is effective (u9JW) and has been robustly evaluated across diverse benchmarks (u9JW, nTe5, PYLD).

Based on the constructive discussions, we have carefully revised the manuscript with the following updates:

- Additional Backbone VLM: New experiments added (Table 5, Sec. 5.3).
- Equation Refinements: Make equations include numbering (Eq. 1 and 2.)
- Clarified Contributions: Despite positive feedback, there were misunderstandings regarding our core focus—pre-training a VLM backbone for optimal VLA representations. We have thus strengthened the Contribution section in the Introduction (Line 95) and Abstract (Line 15).

Revisions in the manuscript are highlighted in $\textbf{\color{blue}blue}$ for convenience. We hope these updates satisfactorily address your concerns.

Thanks for your consideration.

Sincerely,
The Authors

---

### Meta-Review · Area_Chair_23MD · 2026-01-14

**Summary:**

The reviewers were initially mixed on the paper, and raised the following main concerns:

1. Novelty (2 reviewers). The main critique is that the paper is mostly a combination of existing methods (GRPO, supervised fine tuning, tokenization techniques).

2. Limited evaluation (3 reivewers). The original paper focused on one backbone, and the reviewers asked for additional experiments.

3. Comparisons to VLA baselines (2 reviewers). VLAs are a very active area of research, and reviewers asked for other baselines for comparison.

4. Practicality (3 reviewers). There was concern about how practical the method is overall, commenting on improvements in some cases but degradation in other cases.

**Reviewer Concerns:**

The authors did a good job responding to points 2, 3, and 4. They added new experiments with more backbones, and more VLA baselines, which improved the paper. They answered the questions about practicality well, justifying the design choices, and future research could close the remaining gaps.

The concern about novelty is still outstanding, and it is a significant concern. I looked through the paper closely, and I agree with the reviewers that the novelty is limited. The paper is combining many existing methods together, and it is unclear how exciting the contribution is compared to the related VLA work. This is the main concern, and the main reason for the decision. The paper should either expand the novelty, or better explain the contribution and key insights.

**Reviewer Scores:**

I believe the reviewers would have continued to raise concerns about novelty. I expect they would have been satisfied with the other points.

---

### Decision · Program_Chairs · 2026-01-26

Reject